



# Comment on "Exceptionally high heat flux needed to sustain the Northeast Greenland Ice Stream" by S. Smith-Johnson et al., The Cryosphere, 14, 841–854, 2020

Paul D. Bons[1,2], Tamara de Riese[2], Steven Franke[3], Maria-Gema Llorens[4], Till Sachau[2], Nicolas Stoll[3], Ilka Weikusat[2,3], Yu Zhang[2]

[1]China University of Geosciences, Beijing, China.
[2]Department of Geosciences, Eberhard Karls University Tübingen, Tübingen, Germany.
[3]Alfred Wegener Institute Helmholtz Centre for Polar and Marine Research, Bremerhaven, Germany
[4] Geosciences Barcelona, CSIC, Barcelona, Spain

*Correspondence to*: Paul D. Bons (paul.bons@uni-tuebingen.de)

**Abstract.** Smith-Johnson et al. (The Cryosphere, 14, 841–854, 2020) model the effect of a potential hotspot on the Northeast Greenland Ice Stream (NEGIS). They argue that a heat flux of at least 970 mW m$^{-2}$ is required to have initiated or control NEGIS. Such an exceptionally high heat flux would be unique in the world and is incompatible with known geological
processes that can raise the heat flux. NEGIS is thus formed and controlled by some other, yet unknown, process.

## 1 Introduction

The prominent North-East Greenland Ice Stream (NEGIS) is an exceptionally large ice stream in the Greenland Ice sheet. It is over 500 km long, almost reaches the central ice divide and contributes significantly to overall ice drainage from the Greenland Ice sheet (Rignot and Mouginot, 2012; Aschwanden et al., 2016). What causes or drives this ice stream remains
enigmatic. Several authors have suggested that NEGIS was initiated or is controlled by an elevated geothermal heat flux from the underlying bedrock (Fahnestock et al., 2001; Keisling et al., 2014; MacGregor et al., 2016). This hypothesis was investigated in the modelling study of Smith-Johnson et al. (2020). They conclude that *"In our model experiment, a minimum heat flux value of 970 mW m$^{-2}$ located close to the East Greenland Ice-core Project (EGRIP) is required locally to reproduce the observed NEGIS velocities, giving basal melt rates consistent with previous estimates. The value cannot be*
*attributed to geothermal heat flux alone and we suggest hydrothermal circulation as a potential explanation for the high local heat flux"*. The high minimum heat flux of 970 mW m$^{-2}$ mainly derives from Fahnestock et al. (2001), who inferred it from their interpreted 0.1 m a$^{-1}$ melt rate in the upstream area of NEGIS. MacGregor et al. (2016) obtained similar high melting rates in the upstream area of NEGIS, actually in a larger area than that assumed by Smith-Johnson et al. (2020) in their modelling. However, Buchardt and Dahl-Jensen (2007) obtained a more than ten times lower melt rate along the ridge
between GRIP and NorthGRIP, which is on the margin of the high-melting rate area of MacGregor et al. (2016).

## 1 Discussion

An elevated geothermal heat flux is usually attributed to the trail of the Iceland plume (Rogozhina et al., 2016; Martos et al., 2018; Artemieva, 2019). For example, Rogozhina et al. (2016) suggest that the Iceland hotspot left a 400-km-wide, roughly NW to SE oriented swath of elevated geothermal heat flux across Greenland as the crust there was positioned above the
hotspot 35-80 Myr ago. However, the elevated geothermal heat flux in the trail only reaches values in the order of 100 mW m$^{-2}$ and is not expected to have local spikes. Viscous fingering of hot asthenosphere from the Iceland hotspot can potentially heat the overlying crust as far away as the North Sea according to Schoonman et al. (2020). However, temperatures drop off





away from the central Iceland hotspot, especially underneath Greenland, as shown by, for example, the temperature at 80-150 km depth beneath Iceland and the adjacent Atlantic Ocean (Fig. 8 in Lebedev et al., 2017).

Fahnestock et al. (2001) base their inferred high heat flux on the depths of stratigraphic ice layers up to 9000 yr in age, suggesting that the heat flux has at least been so high for the last few thousands of years. A steady-state 970 mW m$^{-2}$ heat flux would imply a local geothermal gradient close to a staggering ca. 400 °C km$^{-1}$ at which felsic rocks would melt at about 2 km depth. Although Fahnestock et al. (2001) suggest that the local bedrock topography is consistent with volcanism, there is no independent evidence for volcanism that is expected above such shallow melting.

Fahnestock et al. (2001) already note that 970 mW m$^{-2}$ is many times the background median value of about 60 mW m$^{-2}$ in continental crust, in which worldwide geothermal heat flux values rarely exceed 200 mW m$^{-2}$ (Hofmeister and Criss, 2005; Davies, 2013). Recently, Rezvanbehbahani et al. (2017) used a machine learning technique that includes tectonic setting, regional geology and ice core measurements to predict a geothermal heat flux in a range of 20-150 mWm$^{-2}$ across Greenland. These values are in line with geothermal heat flux values determined for Antarctica (Dziadek et al. 2017; Burton-Johnson et

al., 2020a,b; Shen et al., 2020), with only local excursions above 200 mW m$^{-2}$ in the tectonically active West Antarctic Rift System (Schroeder et al., 2014).

The geothermal heat flux map of Iceland (Jóhannesson et al. 2020) shows tens of km-sized patches with >200 mW m$^{-2}$ and one smaller spot with 300-350 mW m$^{-2}$, still far below 970 mW m$^{-2}$. Similarly, Yellowstone, which is one of the most active continental hotspots, shows a geothermal heat flux just exceeding 150 mW m$^{-2}$ (Blackwell and Richards, 2004). These two

very active hotspots with active volcanic activity thus have geothermal heat fluxes well below 970 mW m$^{-2}$. If geothermal heat flux values in the Iceland hotspot are <350 mW m$^{-2}$, it is highly unlikely that higher heat fluxes are encountered in its trail.

Active hotspots, such as Iceland and Yellowstone are characterised by volcanic activity that implies the presence of magma chambers or shallow intrusions. Smith-Johnson et al. (2020), recognising that 970 mW m$^{-2}$ is unrealistically high for a

geothermal heat flux, suggest several potential alternative processes that may enhance the high heat flux, such as shallow intrusions. This is in line with Stevens et al. (2016), who conclude, regarding melt, that *"ice-age cycling could help it migrate upward to shallow depth or erupt, contributing to the high observed geothermal heat flux"*, but with the caveat *"if melt occurs at depth"*. The conclusion is based on the vug-wave magma-transport model of Morgan and Holtzman (2005), which is similar to the mobile-hydrofracture transport model of Bons (2001) and Bons et al. (2001). Magma transport in vug

waves or mobile hydrofractures may be enhanced by ice-age cycling or tectonic events, but this will only have effect if magma is present in the source region. The question remains if and why this would be the case underneath the upstream area of NEGIS. Furthermore, the same magma-transport mechanism also applies to igneous activity in hotspots such as Iceland. If the geothermal heat flux there is only raised locally to <350 mW m$^{-2}$, it is unlikely that it would be raised three times more in the Greenland crust where there is no obvious evidence or reason for significant igneous activity.

Another potential cause for the high heat flux that is invoked by Smith-Johnson et al. (2020) (and others e.g., Artemieva, 2019) is hydrothermal fluid flow, which is the flux of hot fluids from deeper levels in the crust that typically leave mineral deposits (Oliver et al., 2006). An indication of the fluid flux required to achieve 0.1 m/yr basal melting can be obtained by assuming that the melting is achieved by 100 °C aqueous fluids that melt basal ice at 0 °C while themselves cooling down to 0 °C. Using a heat capacity of 4.2 kJ kg$^{-1}$ K$^{-1}$ and a latent heat of 334 kJ kg$^{-1}$ for melting ice, we obtain a required fluid flux

of ~2·10$^{-6}$ kg m$^{-2}$ s$^{-1}$ (or ~0.07 m$^3$ m$^{-2}$ yr$^{-1}$). This is more than three orders of magnitude more than the 2-7·10$^{-10}$ kg m$^{-2}$ s$^{-1}$ expected for metamorphic fluid fluxes (Connolly and Thompson, 1989) that could potentially provide the hot fluids. Even the much lower estimated melting rate of 6.1 mm/yr of Buchardt and Dahl-Jensen (2007) would require >10 times the mass of hot fluid than expected. Hydrothermal fluid flow can therefore not produce all the heat required for a significantly elevated basal melting rate.

Uranium enrichments are known in southern Greenland in the Gardar Province (e.g., Bartels et al., 2016), and their



radiogenic heat production can add to the geothermal heat flux directly, and indirectly through enhanced hydrothermal fluid flow, as is the case in the uranium-rich Mt. Painter Inlier in South Australia (Weisheit et al., 2013) where the geothermal heat flux is raised to about 120 mW m$^{-2}$ (Sandiford et al., 1998). In the sediments above the world's largest known U-deposit, Olympic Dam in South Australia, the geothermal heat flux is raised by only 43 mW m$^{-2}$ from a background value of 73 mW
m$^{-2}$ (Houseman et al., 1989).

## 3 Conclusions

In summary, a heat flux of 970 mW m$^{-2}$ is geologically unfeasible. Any heat flux above about 100-150 mW m$^{-2}$ should be treated with caution in the absence of other evidence, such as volcanic or tectonic activity. Most other studies actually do propose much more moderate and realistic geothermal heat flux values below the Greenland Ice sheet (e.g. Buchardt and
Dahl-Jensen, 2007; Rogozhina et al., 2016; Rezvanbehbahani et al., 2017, Artemieva, 2019). The original 970 mW m$^{-2}$ stems from Fahnestock et al. (2001), who derive this value from variations in radar stratigraphy elevation, which they assume to have been caused by basal melting (up to 0.1 m/a). The improbable heat flux-value they derive means that such elevation variations cannot be solely due to basal melting and we need to consider other causes, such as flow heterogeneities in space or time (e.g. due to folding; Bons et al., 2016), as well as the underlying assumptions in determining basal melting.
Even though a ≥970 mW m$^{-2}$ heat flux is here shown to be inhibitively improbable, the statement by Smith-Johnson et al. (2020) that the exceptionally high heat flux is needed to reproduce the observed NEGIS velocities in their model is actually useful. It shows that state-of-the-art simulation codes, such as the sophisticated Ice Sheet System Model (ISSM; Larour et al., 2012; Beyer et al., 2018), apparently miss some critical component(s), as they are not able to replicate a major ice stream such as NEGIS without unrealistic boundary conditions. The studies by both Fahnestock et al. (2001) and Smith-Johnson et
al. (2020) thus highlight the exciting challenge still ahead to truly understand ice streams such as NEGIS and ice-sheet dynamics in general.

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

*Acknowledgements.* M-GL is supported by a Juan de la Cierva-Incorporación (IJC2018-036826-I) fellowship, funded by the Spanish Ministry of Science, Innovation and Universities, NS and IW by the Helmholtz Junior Research group "The effect of deformation mechanisms for ice sheet dynamics" (VH-NG-802), and YZ by the PhD program of the China Scholarship Council (CSC) 202006010063. We thank Nanna Bjørnholt Karlsson for carefully checking the original submission.