# Peer review of "Comment on "Exceptionally high heat flux needed to sustain the Northeast Greenland Ice Stream" by S. Smith-Johnsen et al., The Cryosphere, 14, 841–854, 2020"

_The Cryosphere, 2020_

## Referee Comment (RC1) · Nicholas Holschuh (Referee) · 16 Jan 2021

**Review of:** Comment on "Exceptionally high heat flux needed to sustain the Northeast Greenland Ice Stream" by S. Smith-Johnson et al., The Cryosphere, 14, 841–854, 2020
**Submitted to:** The Cryosphere Discussions
**Reviewer:** Nicholas Holschuh

**General Comments:**
In this comment, the authors evaluate the plausibility of an extreme (970 mW/m$^2$) heat flux anomaly under the Northeast Greenland Ice Stream (NEGIS) by contextualizing the anomaly invoked by Smith-Johnson et al. (2020) with geologic examples elsewhere on Earth. The authors convincingly demonstrate that this value is implausible, and use that conclusion together with the modeling study by Smith-Johnson et al. to say that there must be critical processes or ice sheet characteristics missing in existing ice sheet models of NEGIS.

This comment's discussion of the reasonable bounds on geothermal flux is an incredibly valuable contribution to the literature. I think that the glaciology community has not spent enough time critically evaluating the published values for geothermal flux across both ice sheets. Back of the envelope calculations for both (a) the required volume of hydrothermal fluids and (b) geothermal gradients that would exist for these extreme heat fluxes provide a succinct description of just how extraordinary claims made in the literature can be.

While the authors of this comment provide convincing evidence that a 970 mW/m$^2$ heat flux anomaly does not exist under NEGIS, I do not believe that the original work by Smith-Johnson et al. proves (nor does it state) that an anomaly of that magnitude is *required* to reproduce NEGIS. What it does state is that an extreme anomaly is required to reproduce NEGIS *in their model*, which is contingent on a wide array of assumptions that underpin the specific model experiment. A different set of assumptions would have allowed NEGIS to form, even within their model framework, with lower melt-water input (see the Technical Comments below). Thus, I think some of the language in this comment regarding NEGIS as a system needs to be modified or removed. Once the language in the abstract and conclusions is softened (see the Line-Item Comments), I would be happy to recommend this comment for publication.

**Technical Comments:**
The material properties that describe the NEGIS system, including the transmissivity of the water system at the ice-sediment interface and the effective viscosity of the ice (most notably in the shear margins), are unknown – yet the experiment Smith-Johnson et al. designed treats the geothermal flux as the only free parameter. Thus, the question being asked in the original manuscript is not "what is the minimum GHF that could result in fast flow at NEGIS", but rather "what is the minimum GHF that could result in fast flow at NEGIS *absent any other spatially heterogeneous boundary conditions".* Less heat would be required if the bed were uniformly weaker, if the authors included fabric evolution or imposed viscosity transitions in the margins, or if the water transmissivity at the bed were lower. This point is made in Smith-Johnson et al., in the final two paragraphs on page 851 and the first (half) paragraph on 852.

By excluding the possibility of extreme basal melt, this comment rightfully calls into question the range of assumptions that underpin the Smith-Johnson et al. model. And ultimately, the authors may well be right, that elevated heat flux plays no role in the fast ice flow in Northeast Greenland. But there is no

evidence presented here that can exclude more nuanced explanations that are qualitatively consistent with existing mechanisms: e.g., elevated but reasonable melt rates, combined with a subglacial hydrologic system with very poor conductivity, and a weak till substrate could very well still explain the pattern of ice flow observed. It would be flawed reasoning to say that "by rejecting the Smith-Johnson model of NEGIS, we can reject the possibility that spatial heterogeneity in the geothermal flux plays a role (and potentially, critical role) in the formation of NEGIS."

**Line-Item Comments:**

| | |
|---|---|
| Page #: 1
Line #: 13-15 | On line 13, you overstate the strength of the Smith-Johnson et al. conclusions. A careful read of that manuscript shows they intentionally and repeatedly acknowledge that the high melt rates are required for their model, not required for the NEGIS system, which is an important distinction.  Quotes from the original paper:

*By testing with four mantle plume configurations of increasing intensity (Fig. 2), we find that the GHF (GHF) needed to induce the observed upstream velocity of NEGIS **in our model** is ∼ 970 mW m−2 .*

*This shows that GHF values of 677, 836 and 909 mW m−2 produce weaker ice stream signatures than observed and, **given our model set-up**, are not sufficient to induce the upstream fast flow of NEGIS.*

*The experiments in Fig. 3 indicate that the elevated heat required to initiate the NEGIS **in our model** must be located close to EGRIP.*

***We acknowledge that this value may be overestimated due to uncertainties and assumptions in our model set-up**, and we discuss these in the following sections.*

Ultimately, the work of Smith-Johnson et al. is not enough for you to conclude that NEGIS does not require elevated geothermal flux. You can, however, conclude that the melt rates must be lower than those proscribed in Smith-Johnson et al. This could be fixed with a simple change to line 15: "Thus, fast flow at NEGIS must be possible without the extraordinary melt rates invoked in Smith-Johnson et al." |
| Page #: 1
Line #: 22 | This quotation contains the operative words: "in our model experiment". It is important that proper emphasis is applied to that aspect of their conclusions. |
| Page #: 2
Line #: 50 | Even the Schroeder et al. paper is not a direct observation (and has values pinned to values from an ice sheet model), so this could easily be another example where glaciological modeling overestimates the geothermal flux! |

| | |
|---|---|
| Page #: 3
Line #: 87-89 | The first sentence is perfect -- this should be the key takeaway message of the comment. In additoin, the following few sentences (through line 94) do a good job of motivating why the point made above is important. |
| Page #: 3
Line #: 95-99 | This section, mirroring the abstract, overstates the strength of the conclusions in Smith-Johnson et al. Modifying the language to capture the nuance of the original paper is important. Maybe something along the lines of:

"Given that the extroardinary heat flux invoked in Smith-Johnson et al. (2020) cannot exist at NEGIS, there must exist some other weakness in the NEGIS system that enables fast flow that is not captured by their model. While we cannot rule out a supporting role for geothermal flux at NEGIS, the flux required to produce extreme basal melt invoked by Fahnestock et al., and Smith-Johnson et al. is geologically implausible, leaving open many questions about the dynamics of the NEGIS system. |

---

## Short Comment (SC1) · Dear Bons et al. · 26 Jan 2021

On behalf of my co-authors and myself, we would like to make a general remark about your comment. We agree with your conclusions that the extremely high geothermal heat flux needed to sustain NEGIS in our model is geologically unfeasible. We appreciate your effort to put the 970mW/m2 value into context, providing a detailed background and explaining how unrealistic it is. However, we think you might have overlooked our

manuscript and not appreciated the extent of the limitations we detail in this manuscript. We are very aware of the extremely high value such a geothermal flux represent and thus repeatedly stated this in our paper, starting in the abstract: 'The value cannot be attributed to geothermal heat flux alone and we suggest hydrothermal circulation as a potential explanation for the high local heat flux.' . We deliberately changed the language from the first draft (after valuable comments from Nicholas Holschuh) specifically to underline that this is a model experiment. Another study we performed shows that NEGIS can be reproduced in ISSM with a much lower geothermal heat by for example reducing basal friction within reasonable bounds (e.g. Smith-Johnsen et al. 2020). We therefore do not agree that our study shows that there is a critical component missing in ISSM, as you state in your conclusion. Hopefully the base of EastGRIP will be reached in the near future to provide more constraints on heat flux and basal melt, aiding us to understand the drivers of NEGIS and the role of geothermal flux. I would also like to mention that my name is misspelled throughout your comment (Smith-Johnsen not Smith-Johnson).

Cheers

Silje Smith-Johnsen

Smith-Johnsen, S., Schlegel, N.-J., de Fleurian, B., and Nisancioglu, K.: Sensitivity of the Northeast Greenland Ice Stream to Geothermal Heat, J. Geophys. Res.-Earth, 125, e2019JF005252, https://doi.org/10.1029/2019JF005252, 2020.

---

## Referee Comment (RC2) · Jörg Ebbing (Referee) · 29 Jan 2021

Review of: Comment on "Exceptionally high heat flux needed to sustain the Northeast Greenland Ice Stream" by S. Smith-Johnsen et al., TheCryosphere, 14, 841–854, 2020 Bons et al. comment on the exceptionally high heat flux used in a modelling study by Smith-Johnsen et al. in order to explain the Northeast Greenland Ice Stream. This comment is a valuable contribution as it puts the modelling values used in the first study in perspective and explains that such a high value must be unrealistic, at least on a regional scale without any geological evidence pointing otherwise. The only change,

[Figure]

I would suggest to the manuscript is to revise the statement that "state-of-the-art sim-
ulation codes, such as the sophisticated Ice Sheet System Model (ISSM; Larour et al.,
2012; Beyer et al., 2018), apparently miss some critical component(s), as they are not
able to replicate a major ice stream such as NEGIS without unrealistic boundary con-
ditions." The reason for suggesting to rephrase this sentence is the second study by
Smith-Johnsen et al. (2020) that explores different heat flux models and not necessar-
ily demands an exceptional high heat-flux. Indeed, the uncertainty of the input/model
parameters is a critical element as the comment confirms. Surface heat flux is critical,
but the geophysical models can only provide a rough proxy. Local geological condi-
tions (e.g. bedrock permeability) and topographic gradients affect groundwater flow
and can disturb the effective heat flux from geophysical data sensitive to the deeper
Earth structure. While this will not lead to an equivalent of 970 mW/m**2, this can al-
ter temperatures under the ice, even in a geological old region (see Maystrenko et al.
2015). Such studies, as the comment and the studies by Smith-Johnsen et al. show
that further efforts have to be made to combine geophysical and glaciological mod-
els in a more consistent and realistic manner in order to avoid biases by the applied
parameters.

Maystrenko, Y., Olesen, O. & K. Elvebakk, H. 2015: Indication of deep groundwater
flow through the crystalline rocks of southern Norway. Geology 43, 327–330.

---

## Author Comment (AC1) · 6 Feb 2021

Reply to comments on "Comment on "Exceptionally high heat flux needed to sustain the Northeast Greenland Ice Stream" by S. Smith-Johnsen et al., The Cryosphere, 14, 841–854, 2020"

We thank the two reviewers for their encouraging remarks regarding our comment and their suggestions to improve the manuscript. We also sincerely apologise that neither we, nor the reviewers and editorial staff of The Cryosphere noticed the spelling mistake

in the first-author's name. The reviewers and the authors themselves all point out two issues with our comment:

1) The authors also published another paper (Smith-Johnsen et al., 2019 - actually 2020) that explored the effect of variation in geothermal het flux (GHF) on the modelled ice flow at NEGIS. Reply: This paper was submitted before Smith-Johnsen et al. (2020a) and is referred to in paper under consideration. Smith-Johnsen et al. (2020b) only considered GHF-scenarios with a maximum of about 135 mW/m2 (Greve, 2020b). As we write in our comment, GHF values below ca. 150 mW/m2 are geologically feasible. In the discussion section of Smith-Johnsen et al. (2020b) the authors refer to higher GHF-values "We use five GHF maps to define the uncertainty bounds in the sampling studies, however, GHF values 10 times higher have been suggested for the NEGIS region (Fahnestock et al., 2001). These were not included here, as they are local findings and not spatially distributed maps, and by excluding these high values, we underestimate the ice flux uncertainties". The authors do not write that these values are unrealistic, only that these would increase the uncertainty.

2) The paper by Smith-Johnsen et al. (2020a) does point out at various points in the text that it is a modelling study and the authors also mention that the geothermal heat flux (GHF) is "exceptionally" high. Reply: In our comment, when citing the authors, we did include the caveat "In our model experiment, ...". At the very end of the conclusions, the authors write: "Hence, the minimal heat flux value needed to initiate the ice stream in our model is 970mW/m2, as proposed by Fahnestock et al. (2001). This magnitude is too high to be explained by GHF alone, and we suggest that processes such as hydrothermal circulation may locally elevate the heat flux of the area". This final conclusion does not convey the message that this is a "mere" modelling study to show what the effect of a very high GHF would be and that the high heat flux is questioned, but instead that the GHF alone cannot provide the necessary heat, and that therefore other processes may be needed instead, such as hydrothermal circulation. In our comment we discuss various processes (including hydrothermal circulation) that

could be invoked to elevate the effective heat flux that reaches the bedrock surface and come to the conclusion that it is geologically unlikely that any could raise the value to anywhere near 970 mW/m2.

Whether the authors are aware of, or even in their paper discussed the issues with very high or exceptional heat fluxes is not the main point here. The title of the Smith-Johnsen et al. (2020) paper conveys a strong, clear and unambiguous message: "Exceptionally high heat flux needed to sustain the Northeast Greenland Ice Stream". It is this "take home message" (without ifs or buts) that we take exception to and we think the reviewers support us in expressing our concerns through our comment.

Changes made:

Throughout: Correct spelling mistake in first author's name; again apologies for that.

Line 15: Reviewer Holschuh: Ultimately, the work of Smith-Johnsen et al. is not enough for you to conclude that NEGIS does not require elevated geothermal flux. You can, however, conclude that the melt rates must be lower than those proscribed in Smith-Johnson et al. This could be fixed with a simple change to line 15: "Thus, fast flow at NEGIS must be possible without the extraordinary melt rates invoked in Smith-Johnson et al." Action taken: Following the suggestion by Reviewer Holschuh, we replaced the original sentence "NEGIS is thus formed and controlled by some other, yet unknown, process" with "Fast flow at NEGIS must thus be possible without the extraordinary melt rates invoked in Smith-Johnsen et al."

Line 22: Reviewer Holschuh: This quotation contains the operative words: "in our model experiment". It is important that proper emphasis is applied to that aspect of their conclusions. Action taken: The original text was: "They conclude that "In our model experiment, a minimum heat flux value of 970 mW m-2 located close to the East Greenland Ice-core Project (EGRIP) is required locally to reproduce the observed NEGIS velocities, giving basal melt rates consistent with previous estimates. The value cannot be attributed to geothermal heat flux alone and we suggest hydrothermal circulation as a potential explanation for the high local heat flux". To emphasise that this is based on a model experiment, we changed this to: "They conclude that "... a minimum heat flux value of 970 mW m-2 located close to the East Greenland Ice-core Project (EGRIP) is required locally to reproduce the observed NEGIS velocities, giving basal melt rates consistent with previous estimates. The value cannot be attributed to geothermal heat flux alone and we suggest hydrothermal circulation as a potential explanation for the high local heat flux". It should be noted that this statement is preceded by the caveat "In our model experiment". "

Line 95-99: Reviewer Holschuh: suggests replacing the last sentences with: "Given that the extraordinary heat flux invoked in Smith-Johnson et al. (2020) cannot exist at NEGIS, there must exist some other weakness in the NEGIS system that enables fast flow that is not captured by their model. While we cannot rule out a supporting role for geothermal flux at NEGIS, the flux required to produce extreme basal melt invoked by Fahnestock et al., and Smith-Johnson et al. is geologically implausible, leaving open many questions about the dynamics of the NEGIS system." Reviewer Ebbing also suggests rewording these sentences. Action taken: We took some of the suggested text from the reviewer and merged that with the original text: "Even though the extraordinary heat flux invoked in Smith-Johnsen et al. (2020) cannot exist at NEGIS, their model results are definitively useful. They indicate that some other weakness exists in the NEGIS system that enables the fast flow, most likely with a supporting role of geologically plausible heat fluxes. The studies by both Fahnestock et al. (2001) and Smith-Johnsen et al. (2020) thus highlight the exciting challenge still ahead to truly understand ice streams such as NEGIS and ice-sheet dynamics in general." Two references were removed accordingly.

Other edits: • Missing reference to Rezvanbehbahani et al. (2017) added to reference list. • Author J. Westhoff was accidently omitted from the author list and is now added.

References for this reply: Greve, R.: Geothermal heat flux distribution for the Greenland ice sheet, derived by combining a global representation and information from deep ice cores, Polar Data J., 3, 22–63, 2019. Smith-Johnsen, S., de Fleurian, B., Schlegel, N., Seroussi, H., and Nisancioglu, K.: Exceptionally high heat flux needed to sustain the Northeast Greenland Ice Stream, The Cryosphere, 14, 841–854, doi:10.5194/tc-14-841-2020, 2020a Smith-Johnsen, S., Schlegel, N.-J., de Fleurian, B., and Nisancioglu, K.: Sensitivity of the Northeast Greenland Ice Stream to Geothermal Heat, J. Geophys. Res.-Earth, 125, e2019JF005252, 2020b
* * *

---

## Author Comment (AC2) · 28 Feb 2021

We already provided replies to the reviews of both reviewers (RC1 & RC2) and the authors (SC1) in our comment that was uploaded on February 6, 2021. We see no need to comment further